# Understanding Hyperdimensional Computing for Parallel Single-Pass Learning

**Tao Yu**[*]
Cornell University
tyu@cs.cornell.edu

**Yichi Zhang**[*]
Cornell University
yz2499@cornell.edu

**Zhiru Zhang**
Cornell University
zhiruz@cornell.edu

**Christopher De Sa**
Cornell University
cdesa@cs.cornell.edu

## Abstract

Hyperdimensional computing (HDC) is an emerging learning paradigm that computes with high dimensional binary vectors. There is an active line of research on HDC in the community of emerging hardware because of its energy efficiency and ultra-low latency—but HDC suffers from low model accuracy, with little theoretical understanding of what limits its performance. We propose a new theoretical analysis of the limits of HDC via a consideration of what similarity matrices can be "expressed" by binary vectors, and we show how the limits of HDC can be approached using random Fourier features (RFF). We extend our analysis to the more general class of vector symbolic architectures (VSA), which compute with high-dimensional vectors (hypervectors) that are not necessarily binary. We propose a new class of VSAs, finite group VSAs, which surpass the limits of HDC. Using representation theory, we characterize which similarity matrices can be "expressed" by finite group VSA hypervectors, and we show how these VSAs can be constructed. Experimental results show that our RFF method and group VSA can both outperform the state-of-the-art HDC model by up to 7.6% while maintaining hardware efficiency. This work aims to inspire a future interest on HDC in the ML community and connect to the hardware community.

## 1 Introduction

Hyperdimensional computing (HDC) is an emerging learning paradigm. Unlike conventional cognitive modeling that computes with real numbers, it computes with high dimensional *binary* vectors, referred to as binary *hypervectors*, the dimension of which is usually at least in the thousands. HDC is brain-inspired as high dimensional representations have two fundamental properties similar to human brains: they are **(1)** distributed and highly parallel; and **(2)** robust to noise and tolerant to component failure [Kanerva, 2009]. On the other hand, the massive parallelism and simple arithmetic project HDC into the scope of energy-efficient and ultra-low-latency computing, especially with the rise of emerging hardware [Imani et al., 2021, 2020, Gupta et al., 2020, Salamat et al., 2019]. As a result, HDC has recently attracted considerable attention from edge applications, e.g., robotics, DNA pattern matching, and health diagnosis, as well as data center applications such as recommendation systems [Mitrokhin et al., 2019, Neubert et al., 2019, Neubert and Schubert, 2021, Kim et al., 2020, Burrello et al., 2019, Guo et al., 2021].

---

[*]Equal Contribution.

The practical deployment of HDC is undermined by its low model accuracy compared to other alternatives, e.g., neural networks (NN). The state-of-the-art HDC model on MNIST has an accuracy of 89% [Chuang et al., 2020]. A two-layer NN, however, can easily achieve 95% [Lecun et al., 1998].

There are two main approaches in the literature to improving HDC. One approach is to increase the hypervector dimension, staying within the classic HDC paradigm and just making the binary vectors longer [Neubert et al., 2019, Schlegel et al., 2021]. An alternative is to increase the complexity of each element in a hypervector, e.g., to floating-point or complex numbers (unit circle in the complex plane) [Plate, 1995, Gallant and Okaywe, 2013, Gayler, 1998, Plate, 1994]: this moves the system into the more general realm of *vector symbolic architecture* (VSA) [Schlegel et al., 2021], which uses high-dimensional vectors with elements that are not necessarily binary (unlike binary HDC). However, these remedies are not based on any theoretical analysis of the limits of HDC, and as a result there is a lack of more-than-empirical understanding of when and how they should be deployed.

In this work, we introduce a new notion of expressivity for any VSA using *similarity matrices*. Given a set of hypervectors $v_1, \ldots, v_n$ in a VSA, each entry $\mathbf{M}_{ij}$ in the similarity matrix $\mathbf{M}$ is defined as the similarity between a pair of hypervectors $v_i$ and $v_j$; the similarity is typically measured by an inner product function. Informally, we propose that a VSA is *more expressive* when it can express (i.e. represent with some set of vectors) a wider class of similarity matrices. Importantly, which $\mathbf{M}$ a VSA can express is independent of the vector dimension $D$: this new notion distinguishes between longer-vector and more-complex-vector approaches.

We show that HDC, with binary hypervectors even in *any* dimension, cannot express as many similarity matrices that a VSA with more complex hypervectors can. Even worse, the current method of initializing the hypervectors in an HDC system further reduces the expressible set, which impedes the success of HDC. This notion of expressivity is closely related to learning ability. We exhibit a simple task where current HDC (of any $D$) is incapable of learning a Bayes optimal classifier, while any other VSA system that can express a particular similarity matrix $\mathbf{M}$ (which HDC cannot express) can learn it through the same procedure.

Based on our analysis, we investigate how we can improve HDC through the lens of similarity matrices. We first propose to improve the initialization of binary hypervectors by employing random Fourier features (RFFs) [Rahimi and Recht, 2007]. This method is different from existing approaches that increase the dimension or complexity of hypervectors. We show that this better initialization via expressing a similarity matrix can already surpass state-of-the-art HDC accuracy on MNIST by 6.4%. We then propose and formally define group VSA, an extended version of HDC where elements in hypervectors are more complex than binary but less than floating-point. Group VSA can further improve the RFF-initialized HDC by 1.2% on MNIST.

Our contributions are as follows:

- We provide a theoretical analysis of the limitations of an HDC system with binary hypervectors.
- We approach the expressivity limit of HDC systems with random Fourier features and empirically evaluate the improvements on standard benchmarks.
- We propose group VSA, which generalizes HDC with more complex elements, expanding the set of expressible similarity matrices while maintaining efficiency.
- We evaluate the performance of group VSA on both conventional HDC tasks and image tasks, and study its efficiency implications by analyzing the circuit depths.

## 2    Related Work

The term HDC was first introduced by Kanerva [2009]. It is also referred to as VSA in some literature [Schlegel et al., 2021], a line of work that does symbolic computing. Binary HDC can be traced back to Binary Spatter Code (BSC) [Kanerva, 1994, Kanerva et al., 1997].

**Model capacity improvement.** There are two main VSA formats other than binary HDC: using floating-point real vectors [Plate, 1995, Gallant and Okaywe, 2013, Gayler, 1998, Gosmann and Eliasmith, 2019] or complex vectors [Plate, 1994]. Typically their model capacities are higher since their individual vector components are more complex. Another way of increasing the model capacity is increasing the vector length [Neubert et al., 2019, Schlegel et al., 2021, Chuang et al., 2020, Frady et al., 2018]. However, it remains unknown when and how we should apply these methods, and

whether they are sufficient to solve a task. Our approach is different as our proposed methods are based on the theoretical analysis of, and are designed to bypass, the limits of binary HDC.

**Hardware implication.** HDC inspires a novel hardware architecture that requires *associative memory* [Hopfield, 1982] where long vectors can be stored and addressed efficiently. It is therefore popularized recently in the emerging in-memory computing community [Imani et al., 2020, Gupta et al., 2020]. In the meantime, the simplicity of HDC arithmetic and the massive parallelism make HDC suitable for tasks that require high energy efficiency and low latency. It has been demonstrated successful on commercial hardware as well [Imani et al., 2021, Salamat et al., 2019, Basaklar et al., 2021]. In this work, we provide an analysis on the circuit depths of HDC and our proposed group VSA.

**Theory.** Understanding HDC from a theoretical perspective is currently limited. Thomas et al. [2020], Frady et al. [2018] presented some theoretical foundations of HDC, introducing the benefit of high-dimensional vectors, hypervector encoding, and the connection between HDC and kernel approximation. Our work instead presents the limits of HDC and how we can bypass it. Frady et al. [2021] propose to generalize VSA/HDC to function space. Our work is different since our proposed group VSA, a generalization of HDC, is still discrete and preserves the hardware efficiency.

# 3 Background on HDC

In this section we introduce basics of HDC: hypervectors, arithmetic, and the learning paradigm. We then present a classical approach of HDC on the popular MNIST database. A more comprehensive introduction is in Ge and Parhi [2020].

**HD Representations.** In HDC, we compute with binary hypervectors in a high-dimensional space referred to as *hyperspace*. Given a random hypervector $v$ in a $10,000$ dimensional space $\{-1, 1\}^{10,000}$, it is well known from the "curse" of dimensionality that most vectors in this hyperspace are nearly orthogonal to $v$ [Kanerva, 2009]. We call such hypervectors *unrelated*. The "curse" provides two intriguing properties for cognitive tasks: (1) independent random hypervectors will be unrelated and so can naturally represent objects that are semantically separate, e.g. letters of the alphabet; (2) two hypervectors $u$ and $v$ that have a high-enough inner-product similarity can be classified as being related (i.e. somehow dependent) with high probability. Classical HDC therefore represents data using binary hypervectors randomly drawn from a hyperspace. HDC computes with hypervectors using a fixed set of primitive operations: similarity, binding, bundling, and permutation.

**Similarity.** A similarity function $\mathcal{S}(u, v)$ measures how close two hypervectors $u, v \in \{-1, 1\}^D$ are. It is typically defined as an inner product function [Frady et al., 2021] $\mathcal{S}(u, v) = \frac{1}{D} \sum_{i=1}^{D} u_i v_i$; this is an affine function of the hamming distance for binary hypervectors [Kanerva, 2009].

**Binding $\otimes$.** The binding operation combines two hypervectors $u, v$ into a new hypervector in the same space that represents them as a pair. For binary $\{-1, +1\}^D$, binding is equivalent to coordinate-wise multiplication, i.e. $(u \otimes v) \in \{-1, +1\}^D$ and $(u \otimes v)_i = u_i v_i$ for all $i \in \{1, \ldots, D\}$. Binding preserves similarity, in the sense that $\mathcal{S}(u \otimes w, v \otimes w) = \mathcal{S}(u, v)$ for any hypervectors $u$, $v$, and $w$; also, if $u$ is highly similar to $v$ and $x$ is highly similar to $y$, then $u \otimes x$ will be highly similar to $v \otimes y$ (although usually less than either constituent pair). Binding is implemented on hardware as an XOR.

**Bundling $\oplus$.** Bundling represents an unordered collection of hypervectors. The bundling operation takes in a set of hypervectors and yields a hypervector that is maximally similar to all of them: it acts as an aggregation of a set of hypervectors. Bundling $v_1, \ldots, v_m \in \{-1, +1\}^D$ yields $(\bigoplus_{k=1}^{m} v_k)_i = \mathrm{sgn}\left(\sum_{k=1}^{m} (v_k)_i\right)$ for all $i \in \{1, \ldots, D\}$. This takes a majority vote at each coordinate of the vector; ties are broken at random. HDC typically leverages bundling to learn a class representative.

**Permutation $\prod$.** The permutation operation is a shuffling of the elements in a hypervector. It can be represented as a multiplication of a permutation matrix $\Pi$. A random permutation on a hypervector yields another hypervector that is unrelated to it. Note that permutation is invertible, meaning that $\Pi^{-1} \Pi v_i = v_i$. It is thus useful for encoding order and position information. In hardware, permutation usually appears as *shifting* since its implementation is efficient.

## 3.1 MNIST as a Case Study

We outline how to use the classic HDC approach on the MNIST digit recognition task for illustration.

**Encoding**. First, 256 basis hypervectors $\{v_0, v_1, \ldots, v_{255}\}$ are independently drawn at random from a hyperspace $\{-1, +1\}^D$: each hypervector $v_i$ represents a pixel intensity $i$. Second, we bind all 784 pixels in an MNIST image by their corresponding hypervectors. Since the binding operation is commutative by definition, but pixels in an image have meaningful relative positions, each pixel hypervector is shifted before joining the encoding to preserve that position information. If the input pixel intensities are $p_0, p_1, \ldots, p_{783}$, then its encoded hypervector $t$ is $v_{p_0} \otimes (\Pi v_{p_1}) \otimes \cdots \otimes (\Pi^{783} v_{p_{783}})$, where $\Pi^j$ denotes shifting a hypervector $j$ times.

**Learning**. Encoding yields a set of training hypervectors $\mathbb{T} = \{t_1, t_2, \ldots, t_{60,000}\}$. To learn, we bundle all the hypervectors that are from the same digit. Concretely, a class centroid $s_c$ is computed by $s_c = \bigoplus_{i|y_i=c} t_i$. Each training image is used only once, making this process *single-pass* learning.

**Inference**. At the inference time, a given test image is encoded through the same procedure. The model outputs the class $c$ with the highest similarity $\mathcal{S}(t_{\text{test}}, s_c)$.

# 4 Similarity Matrices and the Limits of HDC

Traditionally the expressivity of HDC setups is identified with the dimension of the hypervectors $D$. This notion is unhelpful for probing the fundamental limitations of HDC, which do not depend on $D$. In this section, we define a new notion of expressivity, which reveals the limits of HDC.

**Definition 1.** *An HDC (or VSA) system can express a similarity matrix $\mathbf{M} \in \mathbb{R}^{n \times n}$ if for any $\varepsilon > 0$, there exists a $D \in \mathbb{N}$ and $D$-dimensional hypervectors $v_1, v_2, \ldots, v_n$ in the HDC/VSA such that $|\mathbf{M}_{ij} - \mathcal{S}(v_i, v_j)| \leq \varepsilon$ where $\mathcal{S}$ denotes the similarity function of the HDC/VSA.*

Informally, this means that the HDC/VSA can approximate $\mathbf{M}$ arbitrarily well. Limitations on $\mathbf{M}$ we can express correspond to limitations on the similarity relation we can represent on data: if we have some dataset and know how similar each pair of examples should be, whether or not we can represent the similarity accurately with an HDC embedding depends on whether HDC can express the corresponding $\mathbf{M}$. Surprisingly, there are some matrices that an HDC system can never express.

**Lemma 4.1.** *Binary HDC can not express the matrix $\mathbf{M}_{\text{Lemma 4.1}} = \frac{3}{2} I_3 - \frac{1}{2} \mathbf{1}_{3 \times 3}$.*

Our notion of expressivity corresponds to learning ability, we give an example task for which whether a VSA/HDC approach can learn the Bayes-optimal classifier depends on whether it can express $\mathbf{M}_{\text{Lemma 4.1}}$. Consider a supervised learning task with input set $\mathcal{X} = \{0, 1, 2\}$, output label set $\mathcal{Y} = \mathcal{X}$, and source distribution $\mathcal{P}(x, y) := (1/9 + 2p)$ if $x = y$ else $(1/9 - p)$ for some small positive number $p$. We say that a VSA can *learn* this task if there exists a $D$-dimensional encoding of $\mathcal{X}$ in that VSA such that, when the bundling method in Section 3.1 is used on a training set of size $N$ drawn from $\mathcal{P}$, the resulting classifier is the Bayes optimal classifier with arbitrarily high probability as $N$ increases.

**Statement 4.1.** *Binary HDC cannot learn this task. Any VSA (formalized later in Definition 2) that can express $\mathbf{M}_{\text{Lemma 4.1}}$ can learn this task.*

Details on Statement 4.1 are in appendix. This learning task shows that only increasing the dimensionality of hypervectors cannot help learn the correct predictions if unable to express a certain matrix. This implies that our notion of expressivity captures HDC limitations in a way that relates to learning.

**Limitations due to initialization.** So far in this section we have described limitations that are inherent to using binary representations in a VSA. Classical HDC methods are often limited in an additional way: rather than considering arbitrary binary hypervectors, they use hypervectors that are sampled independently at random. In such a system, any hypervector used for an embedding (used to represent an entity) is constructed either by (1) independently sampling a binary hypervector where each entry has some probability $p$ of being 1, or (2) permuting and/or binding some pre-existing hypervectors. Examples of this setup can be found in Burrello et al. [2019], Smith and Stanford [1990], Imani et al. [2019b]. Surprisingly, we show that this approach further restricts the set of similarity matrices that can be expressed in expectation.

**Lemma 4.2.** *Let $u_1, u_2, \ldots, u_K$ be binary vectors sampled coordinate-wise independently at random, where each coordinate of $u_i$ has the same probability $p_i$ of being 1. Let $v_0, v_1,$ and $v_2$ be vectors that result from some composition of binding and permutation operations acting on $u_1, \ldots, u_K$, and let*

$\mathbf{M} \in \mathbb{R}^{3 \times 3}$ *be their similarity matrix, such that* $\mathbf{M}_{ij} = \mathcal{S}(v_i, v_j)$. *Then*

$$\left\| \mathbb{E}[\mathbf{M}] - \begin{pmatrix} 1 & -\frac{1}{3} & -\frac{1}{3} \\ -\frac{1}{3} & 1 & -\frac{1}{3} \\ -\frac{1}{3} & -\frac{1}{3} & 1 \end{pmatrix} \right\|_F \geq \frac{\sqrt{2}}{3},$$

*but this target matrix* can be *expressed by binary HDC.*

## 5 Encoding Hypervectors via RFF

Our analysis using similarity matrices provides a strong motivation for using more principled methods to construct hypervectors. We argue that, if there is some similarity matrix $\mathbf{M}$ we want to achieve, we should directly instantiate hypervectors to match it in expectation.

A natural way to represent a desired similarity matrix $\mathbf{M} \in \mathbb{R}^{n \times n}$ is to project it onto the set of representable matrices of binary vectors, which would correspond to a distribution one could sample from. Unfortunately, this approach is intractable as it would require solving a linear programming problem of size exponential in $n$. Instead, to approach the expressivity limits of binary HDC, we

---

**Algorithm 1** Construct correlated hypervectors

**input:** similarity matrix $\mathbf{M} \in \mathbb{R}^{n \times n}$, dimension $d$
**let** $\hat{\Sigma} = \sin(\frac{\pi}{2}\mathbf{M})$ {elementwise}
**let** $U \Lambda U^T = \hat{\Sigma}$ {symmetric eigendecomposition}
**sample** $X \in \mathbb{R}^{n \times d}$ iid unit Gaussians
**return** $\text{sgn}(U\Lambda_+^{1/2}X)$ {elementwise}

---

propose the following approach, given in Algorithm 1. First, we sample $d$ independent multivariate Gaussians over $\mathbb{R}^n$; Our $n$ HDC vectors of length $d$ are then given by the signs of these Gaussians. The following lemma tells us how to make this produce a desired similarity matrix $\mathbf{M}$.

**Lemma 5.1.** *Suppose $X, Y$ are jointly Gaussian zero-mean unit-variance random variables, then* $\mathbb{E}[\text{sgn}(X)\text{sgn}(Y)] = \frac{2}{\pi}\arcsin(\mathbb{E}[XY])$.

From this lemma, it immediately follows that if the elementwise $\sin$ of $\frac{\pi}{2}\mathbf{M}$ is positive semi-definite, then Algorithm 1 produces hypervectors that, in expectation, exactly achieve $\mathbf{M}$; otherwise, some approximation to $\mathbf{M}$ is produced. It also immediately follows that Algorithm 1 can achieve more similarity matrices than the classical procedure of Lemma 4.2: while that lemma shows that the similarity matrix $\frac{4}{3}I_3 - \frac{1}{3}\mathbf{1}_{3 \times 3}$ cannot be achieved in expectation by classical HDC initialization, Algorithm 1 can achieve it as $\sin(\frac{\pi}{2} \cdot \frac{-1}{3}) = \frac{-1}{2}$ and $\frac{3}{2}I_3 - \frac{1}{2}\mathbf{1}_{3 \times 3}$ is positive semidefinite.

Algorithm 1 gives us more freedom to achieve a wider range of similarity matrices; however, it does not tell us *which* similarity matrix $\mathbf{M}$ to choose for a given task and whether $\sin(\frac{\pi}{2}\mathbf{M})$ is positive semi-definite or not. In this paper, we use the well-known RBF kernel [Vert et al., 2004] to choose the similarity matrix between entities, but any similarity matrix appropriate for a task is applicable.

## 6 Group VSA

So far we have shown how replacing existing initialization methods can approach the limits of binary HDC. However, as Lemma 4.1 shows, binary HDC itself has inherent limits. Other known VSAs, such as the unit cycle VSA [Plate, 1994]—in which the elements are complex numbers of absolute value 1—can surpass these limits. However, this comes with the problem of a continuous space—requiring both approximation and significant hardware complexity overhead compared to binary HDC. In this section, we propose a new class of VSA, *finite group VSA*, which effectively "interpolates" between them so as to bypass the similarity-representation limits of binary HDC without the need for a continuous space.

We start by defining a VSA, and then propose to use group structures for the elements of hypervectors as a different approach to improve the expressivity of VSA. Binary HDC can be considered as a special case of our construction corresponding to the 2-element group.

**Definition 2.** *A group VSA is a tuple* $(G, \mu, \otimes, \mathcal{S}, \oplus)$, *where $G$ is some measurable set of symbols,* $\otimes : G \times G \to G$ *is the binding operator,* $\mathcal{S} : G \times G \to \mathbb{R}$ *is a symmetric similarity operator, and* $\oplus : G^{<\omega} \to G$ *is the bundling operator (which maps a finite sequence of symbols to a symbol). A VSA must have the following properties:*

- *Binding.* $(G, \otimes)$ *is a group, i.e., binding is associative over $G$ and has inverse and identity elements.*

- **Similarity to self.** $\mathcal{S}(x, x) = 1$ for all $x \in G$.
- **Similarity preserved by binding.** $\mathcal{S}(g \otimes x, g \otimes y) = \mathcal{S}(x \otimes g, y \otimes g) = \mathcal{S}(x, y)$ for all $g, x, y \in G$.
- **Similarity extensible to an inner product.** There exists a finite-dimensional Hilbert space $V$ over $\mathbb{R}$ and an embedding $\psi : G \to V$ such that $\mathcal{S}(x, y) = \langle \psi(x), \psi(y) \rangle$ for all $x, y \in G$. Equivalently, $\sum_{i=1}^{n} \sum_{j=1}^{n} c_i c_j \mathcal{S}(x_i, x_j) \geq 0$ for any $x_1, \ldots, x_n \in G$ and scalars $c_1, \ldots, c_n \in \mathbb{R}$.
- **Random vectors are dissimilar to any other vector.** $\mathbb{E}_{g \sim \mathrm{Uniform}(G)}[\mathcal{S}(g, x)] = 0$ for any $x \in G$.
- **Bundling.** Bundling of $x_1, \ldots, x_m \in G$ returns $\bigoplus_{i=1}^{m} x_i = \arg\max_{g \in G} \sum_{i=1}^{m} \mathcal{S}(g, x_i)$ or one of the maxima in case of a tie.

To compute using a VSA of dimension $D$, we use hypervectors in $G^D$, extend binding and bundling to act elementwise on these hypervectors, and extend similarity to compute the average similarity over the dimensions as $\mathcal{S}([x_1, \ldots, x_D], [y_1, \ldots, y_D]) = \frac{1}{D} \sum_{i=1}^{D} \mathcal{S}(x_i, y_i)$. It is easy to see that binary HDC is equivalent to a VSA where $G = \{-1, 1\}$, $\otimes$ is multiplication, and $\mathcal{S}(x, y) = xy$. Similarly, a unit-cycle VSA (i.e., FHRR) has $G = \{z \in \mathbb{C} \mid |z| = 1\}$, $\otimes$ as complex multiplication, and $\mathcal{S}(x, y) = \mathrm{Re}(x^* y)$. Most other schemes called "VSAs" in literature fall under our definition, e.g., Gayler [1998], with few exceptions [Plate, 1995] that violate the group requirement. In order to run efficiently on hardware, we add the following restriction.

**Definition 3.** *A finite group VSA is a VSA where $G$ is finite. That is, $(G, \otimes)$ is a finite group.*

On hardware, finite $G$ allows the VSA elements to be represented exactly and lets the VSA operations be computed exactly. The hardware cost will depend on the size of $G$. In many cases, we would like binding to preserve similarity in a stronger sense than that guaranteed by Definition 2. It is often desirable that the similarity after binding is the product of the similarities before binding, i.e.,

$$\mathcal{S}(x_1 \otimes x_2, y_1 \otimes y_2) = \mathcal{S}(x_1, y_1) \cdot \mathcal{S}(x_2, y_2); \tag{1}$$

this would make $\otimes$ behave like a tensor product space with inner product given by $\mathcal{S}$. This property is particularly important, usually when an object consists of multiple features, we will bind these features so as to derive a representative vector for the object, this property ensures that two objects with multiple pairs of similar features to have similar representative vectors.

Of course this equation is *not* guaranteed to hold for elements of $G$ in general (e.g., when $x_1 \otimes x_2 = y_1 \otimes y_2$); however, most VSAs can approximate this behavior by adding an extra randomization step.

**Definition 4.** *Let $A$ denote the uniform distribution over automorphisms of $(G, \otimes, \mathcal{S})$. Then we say that VSA has the* product property *if for any $x_1, x_2, y_1, y_2 \in G$,*
$\mathbb{E}_{\alpha \sim A}[\mathcal{S}(x_1 \otimes \alpha(x_2), y_1 \otimes \alpha(y_2))] = \mathbb{E}_{\alpha \sim A}[\mathcal{S}(\alpha(x_1) \otimes x_2, \alpha(y_1) \otimes y_2)] = \mathcal{S}(x_1, y_1) \cdot \mathcal{S}(x_2, y_2)$.

It is easy to see that this holds for binary HDC, as the identity map is the only automorphism of $\mathbb{Z}/2\mathbb{Z}$ and (1) holds for binary HDC; this also holds for the unit cycle VSA, where the only automorphisms are the identity ($z \mapsto z$) and the complex conjugate automorphism ($z \mapsto z^*$). Rather than using this transformation directly, if one exists we can ensure this "product property" holds by initializing our hypervectors appropriately: if we sample hypervectors $(x_1, y_1)$ at random with independent entries and independently of $(x_2, y_2)$ and both distributions are invariant under automorphisms, then $\mathbb{E}[\mathcal{S}(x_1 \otimes x_2, y_1 \otimes y_2)] = \mathbb{E}[\mathcal{S}(x_1, y_1)] \cdot \mathbb{E}[\mathcal{S}(x_2, y_2)]$.

## 6.1 Constructing a VSA from a finite group

At first glance, the definition of a group VSA may seem open-ended, offering little guidance as to what the limitations of finite group VSAs may be and how they can be constructed. Surprisingly, we can fully characterize the finite group VSAs through *representation theory*. We start by introducing definitions specialized to finite-dimensional complex representations, before stating the full theorem.

**Definition 5** (James et al. [2001], Fulton and Harris [2013]). *A* representation *of a group $G$ over $\mathbb{C}^n$ is a group homomorphism $\rho$ from $G$ to $\mathbb{C}^{n \times n}$ such that $\rho(gh) = \rho(g)\rho(h)$ for all $g, h \in G$.*

*The* character *of a representation $\rho$ is the function $\chi : G \to \mathbb{C}$ given by $\chi(g) = \mathrm{Tr}(\rho(g))$. The representation (and corresponding character) is said to be* irreducible *if no proper subspace of $\mathbb{C}^n$ is preserved by the group action. The* trivial representation*, had by all groups, is $\rho : G \to \mathbb{C}^{1 \times 1}$ with $\rho(g) = 1$ and $\chi(g) = 1$.*

It is a standard result that each finite group possesses a finite number of irreducible characters equal to the number of conjugacy classes of the group [Serre, 1977, Fulton and Harris, 2013].

**Theorem 6.1.** *Let $(G, \otimes)$ be a finite group, and let $X$ denote the set of its non-trivial irreducible characters. Let $\alpha : X \to \mathbb{R}_{\geq 0}$ be some function that assigns a non-negative weight to each of the characters. Then, if we set $\mathcal{S}$ as $\mathcal{S}(g, h) = \frac{\sum_{\chi \in X} \alpha(\chi) \cdot \mathrm{Re}(\chi(g^{-1} \otimes h))}{\sum_{\chi \in X} \alpha(\chi) \cdot \chi(\mathbf{1})}$, where the inverse and unit $\mathbf{1}$ are those of the group, and define bundling $\oplus$ as given in (2), then $(G, \otimes, \mathcal{S}, \oplus)$ is a finite group VSA. Any finite group VSA can be constructed in this way. If in this construction $\alpha$ is supported on only one character $\chi$, i.e. $\mathcal{S}(g, h) = \mathrm{Re}(\chi(g^{-1} \otimes h))/\chi(\mathbf{1})$, then the VSA will have the product property.*

This construction makes it seem as though finite-group VSAs with the product property may be a restricted subset, which could be less expressive. The following result shows that this is not the case.

**Statement 6.1.** *Let $\mathbf{M}$ be a similarity matrix expressible by a finite group VSA. Then there exists a finite group VSA that has the product property and can also express $\mathbf{M}$.*

## 6.2 Cyclic Group VSA

Most of our work with group VSAs in this paper will use the cyclic group $G = \mathbb{Z}/n\mathbb{Z} = \{0, 1, \cdots, n-1\}$, as Definition 2 indicates, we first provide an embedding $\psi : G \to V$ to a finite-dimensional Hilbert space $V$ over $\mathbb{R}$. Let $\psi(x) = (\cos(2\pi x/n), \sin(2\pi x/n))$.

**Definition 6.** *The standard cyclic group VSA is given by:*

- *The symbol set $G = \mathbb{Z}/n\mathbb{Z} = \{0, 1, \cdots, n-1\}$ with addition modulo $n$ as binding operation $\otimes$.*
- *Similarity is defined as $\mathcal{S}(x, y) = \langle \psi(x), \psi(y) \rangle = \cos(2\pi(x - y)/n)$.*

This cyclic group VSA is in some sense a "subset" of the unit cycle VSA, and as $n$ goes to infinity, it approximates the the unit cycle VSA arbitrarily well [Plate, 1994], serving as an interpolation between the binary HDC and the unit cycle VSA. As a straightforward consequence, any $\mathbf{M}$ that can be expressed by this VSA can also be expressed by the unit cycle VSA. To compute with this VSA, we follow the procedure in Section 6: use hypervectors in $G^D$, and extend similarity, binding and bundling operations accordingly. Similar to the HDC case, we utilize random Fourier features for a better basis hypervectors initialization with minor modifications of Algorithm 1; we replace the sgn function in the last step with the ($n$th) quantile function of a Gaussian so as to map into $G$.

Note that as Theorem 6.1 shows, this setup is not the only VSA over the cyclic group. Indeed, for any distribution $\alpha$ over $\{1, \ldots, n-1\}$, the similarity function $\mathcal{S}(x, y) = \sum_{k=1}^{n-1} \alpha(k) \cos\left(\frac{2\pi(x-y)k}{n}\right)$ would yield a finite group VSA. We focus on the VSA of Definition 6 because it satisfies the product property, and all other VSAs on $G$ that do so are either isomorphic to it or isomorphic to the same construction with a smaller $n$.

## 6.3 Non-Abelian Finite Group VSAs

Our analysis of cyclic group VSAs from the previous section extends naturally to cover all finite Abelian groups (i.e. groups in which $\otimes$ is commutative), since it is a classic result that every finite Abelian group factors as the direct product of cyclic groups. It is natural to ask: what about non-Abelian groups? Because they simplify both representation and computation, it would be convenient if we could restrict our attention to Abelian groups only. Unfortunately, the following two statements together show that non-Abelian groups can be strictly more expressive than Abelian groups.

**Statement 6.2.** *Any similarity matrix $\mathbf{M}$ that can be expressed by a finite Abelian group VSA can be expressed by the unit-cycle VSA ($G = \{z \in C \mid |z| = 1\}$, $x \otimes y = xy$, $\mathcal{S}(x, y) = \mathrm{Re}(x^* y)$).*

**Statement 6.3.** *There exists a similarity matrix $\mathbf{M}$ that can be expressed by a VSA over the (non-Abelian) binary icosahedral group, but not by the unit-cycle VSA (i.e., FHRR).*

Statement 6.2 follows from the standard representation-theoretic result that all irreducible representations of a finite Abelian group are one-dimensional, while Statement 6.3 is proved by direct construction. While these results do show that, non-Abelian finite group VSAs are "more powerful" than Abelian finite group VSAs, the additional complexity needed to unlock this power seems not worthwhile for our applications, where unit-cycle VSA already performs well—so, in our experiments we focus solely on the cyclic group. We leave exploration of non-Abelian VSAs to future work.

Table 1: Comparison on test accuracy of proposed methods to SOTA HDC[†] [Imani et al., 2019a], dynamic HDC[*] [Chuang et al., 2020] and 1-bit RFF perceptron. Dimension of hypervectors is 10,000. 1-Epo: 1-Epoch, 10-Epo: 10-Epoch.

| Dataset | ISOLET | | UCIHAR | | MNIST | | Fashion-MNIST | |
|---|---|---|---|---|---|---|---|---|
| Acc(%) | 1-Epo | 10-Epo | 1-Epo | 10-Epo | 1-Epo | 10-Epo | 1-Epo | 10-Epo |
| Percep. | 82.8 | 90.1 | 69.3 | 91.4 | 94.3 | 94.3 | 79.5 | 79.5 |
| HDC[†] | 85.6 | 91.5 | 87.3 | 95.7 | NA | 89.0 | NA | NA |
| RFF HDC | 90.6 | 94.4 | 93.8 | 95.7 | 95.4 | 95.4 | 83.4 | 84.0 |
| RFF G($2^3$)-VSA | 93.1 | 94.4 | 95.1 | 95.6 | 96.3 | 95.7 | 85.4 | **86.7** |
| RFF G($2^4$)-VSA | **94.4** | **96.0** | **95.5** | **96.6** | **96.5** | **96.6** | **87.4** | 86.5 |

## 7 Learning via SGD Instead of Bundling

Prior works train an HDC model via bundling hypervectors in the same class $\mathbb{T}_c = \{t_i | \text{label}(t_i) = c\}$. This is based on the fundamental assumption about bundling that the class representative $s_c$ is similar to each $t_i$. We find that it is *not always true*, depending on the number of vectors being bundled.

Suppose a set $\mathbb{T}_c$ has $2k + 1$ (avoids a tie) unrelated hypervectors, we can theoretically calculate[2] the expected angle $\theta$ between $s_c$ and a randomly selected hypervector $t_i$: $\theta_{2k+1} = \arccos\left(\binom{2k}{k}/2^{2k}\right)$. This indicates that the class vector learned from bundling will be nearly orthogonal to each hypervector in the class and no longer be its representative as we increase $k$.

As an alternative, we propose to leverage stochastic gradient decent (SGD) to learn a linear classifier (same precision). Take binary HDC as an example, the classifier is a binarized matrix multiplication at inference time, i.e., $O = X \cdot \text{sgn}(W)$, where $X$ is the binary hypervector and $W$ is the weight matrix. During the back propagation, we use the straight-through estimator [Hubara et al., 2016] to approximate the gradient of the sign function: $\partial \text{sgn}(W)/\partial W := 1$ *if* $|W| < 1$ *else* $0$.

The inference cost of an HDC model remains the same as the bundling paradigm since they are both binary. The model is still trained for one or few epochs so the SGD approach incurs minor training overhead. We defer the SGD learning process of group VSAs to Appendix.

## 8 Experiments

**Datasets.** We evaluate the performance of proposed methods on two conventional HDC datasets, ISOLET [Dua and Graff, 2017] and UCIHAR [Anguita et al., 2012]. We also evaluate our method on MNIST and Fashion-MNIST [Xiao et al., 2017], which are more challenging for HDC. ISOLET is a speech recognition dataset where each sample is an audio signal with 617 features. Each feature is in the range of $[-1, 1]$. The dataset has 7719 samples in total. The goal is predicting which letter-name was spoken. UCIHAR is a human activity recognition database, each sample of which contains 561 features collected from smartphone sensors. The features are also in the range of $[-1, 1]$. The database has 10299 samples. The task is predicting which type of activity a human was performing.

**Setups.** For ISOLET and UCIHAR, we quantize the features to 8 bits before encoding. We initialize a 10,000-dimensional basis hypervector for each $\{0, \cdots, 255\}$ feature value, then encode raw inputs as described in Section 5 or 6. During the training stage, we use a learning rate of 0.01 and train classifiers for 10 epochs. We compare RFF-HDC and group VSA of order $2^3$ and $2^4$ with SOTA HDC [Imani et al., 2019a, Chuang et al., 2020] [3] that propose iteratively updating the class vectors through misclassified examples. We also compare HDC to a perceptron [Rosenblatt, 1958] where inputs are 10,000-dimensional binary RFFs generated from raw data. We train on Intel Xeon CPUs.

**Results.** 1- and 10-epoch test accuracies are in Table 1, which yield three key observations:

- **RFF HDC already improves non-trivially over the baseline SOTA HDC.** With basis hypervectors initialized from the similarity matrix constructed from pixel similarities, our method improves

---

[2]The calculation is in appendix.

[3]Hernandez-Cane et al. [2021] seem to have a better result on MNIST in Figure 7, but no concrete numbers are reported.

Table 2: Analysis of circuit-depth complexity of binary HDC and 1-bit RFF perceptron.

| Method | CDC |
|---|---|
| Percep. | $91 + 96 \cdot \log_2 N + \frac{3}{2} \log_2 D \cdot (1 + \log_2 D)$ |
| HDC | $\log_2 N + 1 + \frac{3}{2} \log_2 D \cdot (1 + \log_2 D)$ |
| $G(2^n)$-VSA | $3n \log_2 N + 24 \log_2 D$ |

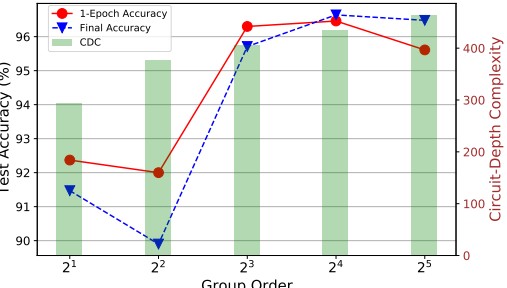

Figure 1: Cyclic Group VSA on MNIST.

the MNIST model accuracy by $6.4\%$ compared to the SOTA. It also for the first time enables HDC learning on Fashion-MNIST, a more challenging task, obtaining 84% final accuracy.

- **Group VSA improves the model accuracy further.** By extending HDC to group VSA, the vector elements are in a higher complexity so that it can more precisely approximate the target similarity matrix than binary hypervectors. Figure 1 shows that when there are 8 or 16 elements in the group, meaning the precision of each element in the hypervector is 3 or 4 bits, the proposed group VSA strikes a good trade-off between accuracy and complexity. It can further outperform RFF HDC by at least 1% across various datasets.

- **Our HDC models learned from a single pass over the data achieve high accuracy.** In all the evaluated tasks, our proposed RFF HDC or the extended group VSA can both achieve approximately the final accuracy in one single epoch. In some cases, e.g., Fashion-MNIST, the group VSA can even obtain a better quality with one single pass. The single-pass model accuracy of HDC is significantly better than the baseline perceptrons, especially on the ISOLET and UCIHAR datasets, which has an at least 8% gap. This evidence shows that HDC learning has an impressive data efficiency. This capability of single-pass learning is consistent with finding in prior works [Hernandez-Cane et al., 2021, Imani et al., 2019a].

## 8.1 Circuit-Depth Complexity

To quantify the potential hardware latency of HDC, we analyze its *circuit-depth complexity* (CDC) in Table 2, defined as the length of the longest path from the input to the output (measured by the number of two-input gates along the path). CDC is commonly used to analyze the complexity of Boolean functions. We further assume that operations without data dependencies are all in parallel. Let $N$ be the feature vector length, e.g., $N = 784$ for an MNIST image, $D$ be the hypervector dimension.

**Binary HDC.** The encoding stage binds all feature hypervectors, which can be implemented in a tree structure. The depth of a single binding operation (XNOR) is $1$. The total depth is therefore $\log_2 N$. Computing the similarity includes a binding and a bit counting. We assume a B-bit ripple carry adder, a chain of full adders, has a depth of $3 \cdot B$ [Satpathy, 2016]. Therefore a D-bit population count has a depth of $3 + 6 + \cdots + 3 \log_2 D = \frac{3}{2} \log_2 D \cdot (1 + \log_2 D)$.

**Cyclic Group VSAs.** For a cyclic group VSA of order $2^n$, the depth of a single binding is $3n$ as it is an addition over the group. Therefore, binding all features has a depth of $3n \log_2 N$. For similarity computations, we precompute the similarity matrix, which consists of $\mathcal{S}(x, y), \forall x, y \in G$ in 8-bit numbers. Hence, the depth of computing the similarity for hypervectors is $3 \cdot 8 \cdot \log_2 D$.

**1-bit RFF perceptron.** Projecting a feature vector onto a selected basis requires a depth of a 32-bit multiplier and an adder. A 32-bit Wallece tree multiplier [Wallace, 1964] has roughly a depth of 45. A 32-bit ripple carry adder has a depth of 96. A cosine operation generating random fourier features requires about the same depth as a 32-bit multiplier if computing with the CORDIC algorithm [Volder, 1959]. Since the perceptron is 1-bit, computing the distance has the same depth as that of HDC.

As a result, binary HDC has a CDC of 295 on MNIST and the cyclic group $G(2^3)$ VSA has 405. The complexity of HDC is $4.4\times$ lower than a 1-bit RFF perceptron with a depth of 1299, while CDC of the cyclic group $G(2^3)$ VSA is $3.2\times$ lower. HDC and group VSA are much faster in potential. In Figure 1, we plot the performance and CDC of a cyclic group VSA when the order varies. Our code is available on github [4].

---

[4] https://github.com/Cornell-RelaxML/Hyperdimensional-Computing

## 8.2 Discussion

The performance of various HDC/VSA methods is closely related to the set of its expressible similarity matrices. As a matter of fact, the required similarity matrix to learn (some) tasks in the paper might already be covered by (or close to in terms of Frobenius norm) the set of expressible similarity matrices of the 10k-dimensional RFF HDC. Hence, the improvement from group-VSA can be limited compared to RFF HDC. If instead considering a 1k-dimensional binary RFF HDC with a smaller set of expressible similarity matrices, group-VSA demonstrates a much better accuracy improvement. For example, on MNIST, 1k-dimensional binary RFF HDC achieves $65.59\%$ 10-epoch test accuracy on MNIST. $G(2^3)$-VSA, meanwhile, achieves $88.61\%$, and $G(2^4)$-VSA achieves $92.56\%$ test accuracy.

The circuit depth serves as a preliminary analysis on the hardware complexity of HDC. While other efficient circuits, e.g., a parallel adder instead of a ripple carry adder, will have lower depth and make HDC attractive further, we avoid being over-optimistic on the estimation. For a practical hardware implementation, better circuits should be applied. Besides, circuit depth only reflects the latency. In the future, an estimation on the number of operations will reflect the energy or circuit area and will further improve the analysis.

## 9   Conclusion

From our theoretical analysis, there is a clear connection between the class of expressible similarity matrices and the expressivity of HDC/VSA. This new notion of expressivity reveals the limits of HDC that computes with binary hypervectors, and meanwhile provides a hint on how we can improve it. The nontrivial improvement from group VSA and the proposed techniques on HDC across various benchmarks suggests that this notion paves a new way towards the future development of HDC/VSA.

## Acknowledgments and Disclosure of Funding

This work is supported in part by NSF Awards IIS-2008102 and CCF-2007832, and by CRISP, one of six centers in JUMP, a Semiconductor Research Corporation program sponsored by DARPA. The authors would like to thank Denis Kleyko from the Redwood Center for Theoretical Neuroscience at UC Berkeley and researchers from VSAONLINE for providing valuable feedbacks on earlier versions of this paper.

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
