# Appendix: Understanding Hyperdimensional Computing for Parallel Single-Pass Learning

## A Proofs of Lemmas, Statements and Theorems

**Lemma 4.1.** *No binary HDC can express the following similarity matrix*

$$\mathbf{M} = \begin{pmatrix} 1 & -\frac{1}{2} & -\frac{1}{2} \\ -\frac{1}{2} & 1 & -\frac{1}{2} \\ -\frac{1}{2} & -\frac{1}{2} & 1 \end{pmatrix} \ .$$

*Proof.* There are $n = 3$ basic entities, where we have some HDC vectors $\boldsymbol{v}_0, \boldsymbol{v}_1, \boldsymbol{v}_2 \in \mathbb{R}^D$ which can be any dimension. We start from $D = 1$ case, with the inner product as the similarity measurement, we can easily enumerate all possible similarity matrices as follows:

$$\begin{pmatrix} 1 & 1 & 1 \\ 1 & 1 & 1 \\ 1 & 1 & 1 \end{pmatrix}, \ \begin{pmatrix} 1 & -1 & 1 \\ -1 & 1 & -1 \\ 1 & -1 & 1 \end{pmatrix}, \ \begin{pmatrix} 1 & 1 & -1 \\ 1 & 1 & -1 \\ -1 & -1 & 1 \end{pmatrix}, \ \begin{pmatrix} 1 & -1 & -1 \\ -1 & 1 & 1 \\ -1 & 1 & 1 \end{pmatrix}.$$

When $D > 1$, note that

$$\mathcal{S}(\boldsymbol{v}_i, \boldsymbol{v}_j) = \sum_{k=1}^{D} \mathcal{S}(\boldsymbol{v}_{ik}, \boldsymbol{v}_{jk})/D \ ,$$

which indicates that all possible similarity matrices must reside in the convex hull of the similarity matrices enumerated above because $D$ can be of any dimension. Easy to verify that this convex hell does not contain $\mathbf{M}$: thus no binary HDC can achieve it. $\qquad\square$

**Statement 4.1.** *Binary HDC cannot learn the following task.*

*Consider a supervised learning task with input example set $\mathcal{X} = \{0, 1, 2\}$, output label set $\mathcal{Y} = \mathcal{X}$, and source distribution*

$$\mathcal{P}(x, y) = \begin{cases} 1/9 + 2p & x = y \\ 1/9 - p & x \neq y \end{cases}$$

*for some small positive number $p$.*

*Proof.* Let $\phi : \mathcal{X} \to \{-1, 1\}^D$ be any binary HDC encoding, and extend $\phi(x) = \phi(x \bmod 3)$ when $x > 3$. Given a class $\hat{y}$, we can then compute the class representative $c_{\hat{y}}$ as

$$\begin{aligned}
\boldsymbol{c}_{\hat{y}} &= \bigoplus_{x : y(x) = \hat{y}} \phi(x) = \mathrm{sgn}(\mathbb{E}[\phi(x)|\hat{y}]) \\
&= \mathrm{sgn}[(\frac{1}{3} - 3p)(\phi(\hat{y} + 1) + \phi(\hat{y} + 2)) + (\frac{1}{3} + 6p)\phi(\hat{y})] \\
&= \mathrm{sgn}[(\frac{1}{3} - 3p)(\phi(\hat{y}) + \phi(\hat{y} + 1) + \phi(\hat{y} + 2)) + 9p\phi(\hat{y})] \ .
\end{aligned}$$

Note that $\phi(\hat{y}) + \phi(\hat{y} + 1) + \phi(\hat{y} + 2)$ cannot be a zero vector; otherwise,

$$\begin{aligned}
0 &= \mathcal{S}(\phi(\hat{y}), \phi(\hat{y}) + \phi(\hat{y} + 1) + \phi(\hat{y} + 2)) \\
&= 1 + \sum_{j \neq \hat{y}} \mathcal{S}(\phi(\hat{y}), \phi(j)),
\end{aligned}$$

due to symmetry, one can easily derive that $\mathcal{S}(\phi(i), \phi(j)) = -1/2$ for $i \neq j$, then $\mathbf{M}$ is achieved, a contradiction to Lemma 4.1.

Since $p$ is small positive number, then the sign of $\mathbb{E}[\phi(x)|\hat{y}]$ is dominated by the first term $\phi(\hat{y}) + \phi(\hat{y}+1) + \phi(\hat{y}+2)$. Hence, the class representative of class $\hat{y}$ is computed as

$$c_{\hat{y}} = \bigoplus_{x:y(x)=\hat{y}} \phi(x) = \text{sgn}(\mathbb{E}[\phi(x)|\hat{y}])$$
$$= \text{sgn}(\phi(\hat{y}) + \phi(\hat{y}+1) + \phi(\hat{y}+2)),$$

which is same for each class, i.e., binary HDC fails to learn this simple task.

On the other hand, for any VSA that can express $\mathbf{M}$ with $\boldsymbol{v}_0, \boldsymbol{v}_1, \boldsymbol{v}_2$, set $\phi(x) = \boldsymbol{v}_x$. We can compute the class representative $c_{\hat{y}}$ as

$$c_{\hat{y}} = \bigoplus_{x:y(x)=\hat{y}} \phi(x) = \arg\max_z \langle z, \mathbb{E}[\phi(x)|\hat{y}]\rangle$$
$$= \arg\max_z \langle z, (\frac{1}{3} - 3p)(\phi(\hat{y}) + \phi(\hat{y}+1) + \phi(\hat{y}+2)) + 9p\phi(\hat{y})\rangle$$
$$= \arg\max_z \langle z, 9p\phi(\hat{y})\rangle = \phi(\hat{y}).$$

The class representative of class $\hat{y}$ will be

$$c_{\hat{y}} = \text{sgn}(\mathbb{E}[\phi(x)|\hat{y}]) = \text{sgn}(\frac{3p}{2}\phi(\hat{y})) = \phi(\hat{y}),$$

which gives a Bayes optimal classifier, outputs the most probable class and proves Statement 4.2. $\square$

**Statement 4.2.** *Any VSA (formalized in Definition 2) that can express $\mathbf{M}_{\text{Lemma 4.1}}$ can learn this task.*

**Lemma 4.2.** *Let $u_1, u_2, \ldots, u_K$ be binary vectors sampled coordinate-wise independently at random, where each coordinate of $u_i$ has the same probability $p_i$ of being 1. Let $v_0$, $v_1$, and $v_2$ be vectors that result from some composition of binding and permutation acting on $u_1, \ldots, u_K$, and let $\mathbf{M} \in \mathbb{R}^{3\times 3}$ be their similarity matrix, such that $\mathbf{M}_{ij} = \mathcal{S}(v_i, v_j)$. Then*

$$\left\| \mathbb{E}[\mathbf{M}] - \begin{pmatrix} 1 & -\frac{1}{3} & -\frac{1}{3} \\ -\frac{1}{3} & 1 & -\frac{1}{3} \\ -\frac{1}{3} & -\frac{1}{3} & 1 \end{pmatrix} \right\|_F \geq \frac{\sqrt{2}}{3},$$

*but this target matrix is expressible by some binary HDC.*

*Proof.* It is straightforward to show that for some $x, y, z \in [-1, 1]$

$$\mathbf{E}[\mathbf{M}] = \begin{pmatrix} 1 & xy & xz \\ xy & 1 & yz \\ xz & yz & 1 \end{pmatrix}.$$

But since $(xy) \cdot (xz) \cdot (yz) = x^2 y^2 z^2$ is a square number, it follows that the upper-triangular elements cannot all be negative. At least one of them must be non-negative, from which the result immediately follows. A binary HDC that achieves this matrix is: $(-1, 1, 1), (1, -1, 1), (1, 1, -1)$. $\square$

**Lemma 5.1.** *Suppose that $X$ and $Y$ are jointly Gaussian zero-mean unit-variance random variables. Then*

$$\mathbb{E}[\text{sgn}(X)\,\text{sgn}(Y)] = \frac{2}{\pi} \arcsin\left(\mathbb{E}[XY]\right)$$

*Proof.* Without loss of generality let $U \sim \mathcal{N}(0, I)$ be a standard Gaussian over $\mathbb{R}^2$, and suppose that $X = a^T U, Y = b^T U$ for some vectors $a, b \in \mathbb{R}^2$ with $\|a\| = \|b\| = 1$ and $a^T b = \mathbb{E}[XY]$. Here, a geometric argument shows that $\mathbb{P}(X \geq 0 \wedge Y \leq 0) = \mathbb{P}(a^T U \geq 0 \wedge b^T U \leq 0) = \theta/(2\pi)$, where $\theta$ is the angle between $a$ and $b$. An analogous analysis of the other three cases, combined with some straightforward trigonometry, proves the lemma. $\square$

**Theorem 6.1.** *Let $(G, \otimes)$ be a finite group, and let $X$ denote the set of its non-trivial irreducible characters. Let $\alpha : X \to \mathbb{R}_{\geq 0}$ be some function that assigns a non-negative weight to each of the characters. Then, if we set $\mathcal{S}$ to be*

$$\mathcal{S}(g, h) = \frac{\sum_{\chi \in X} \alpha(\chi) \cdot \text{Re}(\chi(g^{-1} \otimes h))}{\sum_{\chi \in X} \alpha(\chi) \cdot \chi(\mathbf{1})},$$

*where the inverse and unit $\mathbf{1}$ are those of the group $(G, \otimes)$, and define bundling $\oplus$ as given in the definition of a group VSA, then $(G, \otimes, \mathcal{S}, \oplus)$ is a finite group VSA. Any finite group VSA can be constructed in this way.*

*If in this construction $\alpha$ is supported on only one character $\chi$, i.e. $\mathcal{S}(g, h) = \mathrm{Re}(\chi(g^{-1} \otimes h))/\chi(\mathbf{1})$, then the VSA will have the product property.*

*Proof.* The first part of this theorem is a direct consequence of the following more technically-stated theorem.

The second part follows directly from the fact that if $\phi$ is an irreducible representation of a finite group $G$, and $A$ is the set of automorphisms of $G$, then for any $g \in G$,

$$\frac{1}{|A|} \sum_{a \in A} \phi(a(g)) = cI$$

for some scalar $c$. In particular, this means that if $\chi$ is the corresponding character, then for any $g, h \in G$,

$$\frac{1}{|A|} \sum_{a \in A} \chi(h^{-1} a(g)) = \frac{1}{|A|} \sum_{a \in A} \mathrm{tr}\left(\phi(h^{-1})\phi(a(g))\right)$$

$$= \mathrm{tr}\left(\phi(h^{-1})\frac{1}{|A|} \sum_{a \in A} \phi(a(g))\right)$$

$$= \mathrm{tr}\left(\phi(h^{-1}) \cdot cII\right)$$

$$= c \cdot \chi(h^{-1}).$$

Substituting $h = 1$ yields that $c = \mathrm{Re}(\chi(g))/\chi(1)$ (since $\chi$ must also be preserved by automorphisms up to complex conjugation), which immediately implies what we wanted to prove. $\square$

**Theorem A.1** (Representation theorem for group VSAs). *Suppose that we have a finite group $G$ equipped with a similarity measure denoted $\langle \cdot | \cdot \rangle_G$. Let $X$ denote the set of non-trivial irreducible characters of $G$ (i.e. $X$ is the character table of $G$ excluding the top row). Then there exists a unique function $\alpha : X \to \mathbb{R}_{\geq 0}$ such that $\alpha(\bar{\chi}) = \alpha(\chi)$ for all $\chi \in X$, $\sum_{\chi \in X} \alpha(\chi) \cdot \chi(1) = 1$, and*

$$\langle \cdot | \cdot \rangle_G = \sum_{\chi \in X} \alpha(\chi) \cdot \chi(g^{-1}h). \tag{1}$$

*Conversely, for any function $\alpha$ of this type, if we define $\langle \cdot | \cdot \rangle$ according to (1), then $\langle \cdot | \cdot \rangle$ will be a similarity measure for $G$.*

*Additionally, if we define $M$ as the matrix such that $M_{gh} = \langle g | h \rangle_G$, and $d$ is the rank of $M$, then there exists some positive integer $K$ and positive integers $d_1, d_2, \ldots, d_K$ such that $\sum_{k=1}^{K} d_K = 1$, and there exists a $|G|$-dimensional subspace $\mathcal{A}$ of $\mathbb{R}^{d_1 \times d_1} \times \mathbb{R}^{d_2 \times d_2} \times \cdots \times \mathbb{R}^{d_K \times d_K}$ and function $\phi : G \to \mathcal{A}$ such that for all $g, h \in G$,*

$$\phi(g)\phi(h) = \phi(gh)$$
$$\phi(g)^{-1} = \phi(g)^T = \phi(g^{-1})$$
$$\phi(1) = \phi(I),$$

*where multiplication and transposition in $\mathcal{A}$ is done component-wise on the $K$ components (each of which is a matrix), and such multiplication preserves $\mathcal{A}$. Also, there exist some positive scalars $\beta_1, \beta_2, \ldots, \beta_K$ such that if we define an inner product on $\mathcal{A}$ as*

$$\langle (x_1, x_2, \ldots, x_K), (y_1, y_2, \ldots, y_K) \rangle_{\mathcal{A}} = \sum_{k=1}^{K} \beta_k \, \mathrm{tr}\left(x_k^T y_k\right),$$

*where here each $x_k \in \mathbb{R}^{d_k \times d_k}$, then*

$$\langle g | h \rangle_G = \langle \phi(g) | \phi(h) \rangle_{\mathcal{A}}.$$

*Proof.* Let $M$ be the matrix described in the theorem statement, such that $M_{gh} = \langle g|h \rangle_G$. Observe that $M$ must be symmetric and positive semidefinite (by the properties of the similarity measure). For some $f \in G$, let $B_f$ denote the matrix

$$B_f = \sum_{g \in G} e_{fg} e_g^T,$$

where $e_g$ is the unit basis element associated with $g$. Observe that $B_f^T = B_f^{-1} = B_{f^{-1}}$, and that $B_f$ commutes with $M$, because

$$
\begin{aligned}
e_h^T M B_f e_g &= e_h^T M e_{fg} \\
&= \rho(h^{-1} fg) \\
&= \rho((f^{-1}h)^{-1} g) \\
&= e_{f^{-1}h}^T M e_g \\
&= \left( B_{f^{-1}} e_h \right)^T M e_g \\
&= e_h^T B_{f^{-1}}^T M e_g \\
&= e_h^T B_f M e_g.
\end{aligned}
$$

That is, $MB_f = B_f M$. Since $B_f$ commutes with $M$, it also must commute with any polynomial of $M$. In particular, if the nonzero eigendecomposition of $M$ is

$$M = \sum_{k=1}^{K} \lambda_k V_k,$$

where each $\lambda_k > 0$ is distinct, and $V_k^2 = V_k$ is a symmetric projection matrix onto the associated eigenspace, then since $V_k$ can be expressed as a polynomial in $M$, it must also commute with $B_f$ for all $f$. The same thing will be true for $C_f$ defined as

$$C_f = \sum_{g \in G} e_{gf} e_g^T.$$

These results together show that for any $f, g, h \in G$,

$$e_h^T V_k e_g = e_{fh}^T V_k e_{fg} = e_{hf}^T V_k e_{gf}.$$

Now, let $d_k$ denote the rank of $V_k$ (the multiplicity of the eigenvalue $\lambda_k$ in $M$). So, there must exist some matrix $W_k \in \mathbb{R}^{|G| \times d_k}$ such that $V_k = W_k W_k^T$ and $W_k^T W_k = I$. For any $g \in G$, let $U_k(g)$ denote the matrix in $\mathbb{R}^{d_k \times d_k}$

$$U_k(g) = W_k^T B_g W_k.$$

Observe that $U_k(1) = I$, all the $U_k(g)$ matrices are orthogonal, and

$$
\begin{aligned}
U_k(g) U_k(h) &= W_k^T B_g W_k W_k^T B_h W_k \\
&= W_k^T B_g V_k B_h W_k \\
&= W_k^T B_g B_h V_k W_k \\
&= W_k^T B_{gh} W_k \\
&= U_k(gh).
\end{aligned}
$$

That is, $U_k$ is a representation of the group $G$. Observe that the trace of $U_k(g)$ is

$$
\begin{aligned}
\operatorname{tr}\left(U_k(g)\right) &= \operatorname{tr}\left(W_k^T B_g W_k\right) \\
&= \operatorname{tr}\left(W_k W_k^T B_g\right) \\
&= \operatorname{tr}\left(V_k B_g\right) \\
&= \sum_{h \in G} e_h^T V_k B_g e_h \\
&= \sum_{h \in G} e_h^T V_k e_{gh} \\
&= \sum_{h \in G} e_1^T V_k e_g \\
&= |G| \cdot e_1^T V_k e_g.
\end{aligned}
$$

In particular, this means that $g \mapsto |G| \cdot e_1^T V_k e_g$ is a character of $G$. As a character, it must be a sum of irreducible characters of $G$, and since it is real, it must place the same weight on complex-conjugate characters. It follows that $g \mapsto \sum_{k=1}^K \lambda_k e_1^T V_k e_g$ must be a non-negative scaled sum of irreducible characters of $G$ that places the same weight on complex-conjugate characters. But, this function is just $g \mapsto e_1^T M e_g$, which is just $\rho$. So, $\rho$ must be a non-negative sum of the irreducible characters of $G$. The fact that this scaling is unique follows from the fact that the characters are linearly independent; the fact that this scaling only contains non-trivial characters follows from the average-dissimilarity property.

We then construct an algebra $\mathcal{A}$ as follows. Let $\phi(g) : G \to \mathbb{R}^{d_1 \times d_1} \times \cdots \times R_{d_K \times d_K}$ be defined as

$$
\phi(g) = (U_1(g), U_2(g), \ldots, U_K(g)),
$$

and let the inner product scalars be

$$
\beta_k = \frac{\lambda_k}{|G|}.
$$

Then

$$
\begin{aligned}
\sum_{k=1}^K \beta_k \operatorname{tr}\left(U_k(h)^T U_k(g)\right) &= \sum_{k=1}^K \beta_k \operatorname{tr}\left(U_k(h^{-1})U_k(g)\right) \\
&= \sum_{k=1}^K \beta_k \operatorname{tr}\left(U_k(h^{-1}g)\right) \\
&= e_1^T M e_{h^{-1}g} \\
&= \rho(h^{-1}g) \\
&= \langle h | g \rangle_G
\end{aligned}
$$

as desired. To finish the construction, let $\mathcal{A}$ be the algebra spanned by $\{\phi(g) \mid g \in G\}$. Observe that this must be closed under multiplication and transposition because $G$ is closed under multiplication and inversion. $\qquad \square$

**Statement 6.1.** *Let $\mathbf{M}$ be similarity matrices expressible by a finite group VSA. Then there exists a finite group VSA that has the product property and can also achieve $\mathbf{M}$.*

*Proof.* Suppose the first VSA's group is $G$ and has irreducible characters $\chi_1, \chi_2, \ldots, \chi_k$. Then the group $G^k$ consisting of the direct product of $k$ copies of the group $G$, together with a similarity function $\mathcal{S}((x_1, \ldots, x_k), (y_1, \ldots, y_k)) \propto \prod_{i=1}^k \chi_i(x_i^{-1} \otimes y_i)$ will both have the product property (as its similarity matrix is proportional to a single character) and can express any similarity matrix the first VSA can. $\qquad \square$

**Statement 6.2.** *Any similarity matrix $\mathbf{M}$ that can be expressed by a finite Abelian group VSA can be expressed by the unit-cycle VSA ($G = \{z \in C \mid |z| = 1\}$, $x \otimes y = xy$, $\mathcal{S}(x, y) = \operatorname{Re}(x^* y)$).*

*Proof.* Note the fact that all irreducible representations of finite Abelian groups are 1-dimensional. Then the results follows immediately with any irreducible representation as the mapping from a finite Abelian group to $G = \{z \in C \mid |z| = 1\}$. $\qquad\square$

**Statement 6.3.** *There exists a similarity matrix* $\mathbf{M}$ *that can be expressed by a VSA over the (non-Abelian) binary icosahedral group, but not by the unit-cycle VSA.*

*Proof.* Consider the binary icosahedral group expressed as a subset of the quaternions. This group consists of 120 elements which are placed at the vertices of a 600-cell inscribed in the unit 3-sphere. Consider an arbitrary sequence of 60 of these elements containing exactly one of $\{x, -x\}$ for each $x$ in the binary icosahedral group: i.e. we select exactly one of each pair of antipodal points in the group. This sequence of 60 points has a similarity matrix $M \in \mathbb{R}^{60 \times 60}$. It is easy to check that the absolute values of the entries of this matrix lie in $\{0, \frac{1}{2\phi}, \frac{1}{2}, \frac{\phi}{2}, 1\}$, where $\phi = \frac{1+\sqrt{5}}{2}$ is the golden ratio. Define the matrix $A \in \mathbb{R}^{60 \times 60}$ such that $A_{ij} = \pm 1$ if $M_{ij} = \pm\frac{\phi}{2}$ and $A_{ij} = 0$ otherwise. Consider the optimization problem to maximize $\text{tr}(AZ)$ over positive semidefinite matrices $Z \in \mathbb{R}^{60 \times 60}$ subject to the constraint that the diagonal of $Z$ is all-ones, i.e. $Z_{ii} = 1$. It is easy to check numerically that the only solution to this optimization problem is $Z = M$. Also observe that $M$ has rank greater than 2.

Now, suppose that there existed some representation of $M$ using vectors with entries in the unit circle in $\mathbb{C}$. For this to hold, it would need to be the case that $M$ is in the convex combination of the similarity matrices generated by those entries, each of which must be of rank 2. But, this is impossible, since (1) none of those matrices can be equal to $M$, and as such (2) any such matrix $M_C$ will have $\text{tr}(M_C A) < \text{tr}(MA)$. This shows that this particular matrix can't be represented by hypervectors with unit-absolute-value entries in $\mathbb{C}$. $\qquad\square$

# B   Calculation of $\theta$ in Section 7

We can theoretically calculate the angle $\theta$ between the class vector $s_c$ and a randomly selected hypervector $t_i$ in the set $\mathbb{T}_c$.

$$\cos\theta = \frac{s_c \cdot v_j}{\|s_c\| \, \|v_j\|} = \frac{2 \cdot q - D}{D}$$

$D$ is the dimension of $t_i$. $q$ is the number of elements that have the same sign in $s_c$ and $t_i$. Since

$$s_c = \text{sgn}\left(\bigoplus_{j \in \mathbb{T}_c} t_j\right) = \text{sgn}\left(t_i + \sum_{j=1, j \neq i}^{2k+1} t_j\right),$$

$q$ is proportional to the probability $p_k$ of the sign of an entry in $t_i$ will be flipped after adding the term $\sum_{j=1, j \neq i}^{2k+1} t_j$ (the other $2k$ vectors) to it. Obviously $q = D \cdot p_k$ as each entry in $t_i$ is independent.

In order to avoid flipping the sign of an entry, there should be at least $k$ entries out of $2k$ that have the same sign, so the probability $p_k$ can be calculated by

$$p_k = \frac{\binom{2k}{k} + \binom{2k}{k+1} + \cdots + \binom{2k}{2k}}{2^{2k}} = \frac{1 + \frac{1}{2^{2k}}\binom{2k}{k}}{2}$$

Plug the expression of $p_k$ into $\cos\theta$, we can get

$$\theta_{2k+1} = \cos^{-1}\left(\frac{1}{2^{2k}} \cdot \binom{2k}{k}\right)$$

Importantly, $p_k$ is monotonic.

*Proof.*

$$p_{k+1} = \frac{1 + \frac{1}{2^{2k}} \cdot \frac{1}{4} \cdot \binom{2k+2}{k+1}}{2}$$

$$= \frac{1 + \frac{1}{2^{2k}} \cdot \frac{1}{4} \cdot [\binom{2k+1}{k+1} + \binom{2k+1}{k}]}{2}$$

$$= \frac{1 + \frac{1}{2^{2k}} \cdot \frac{1}{4} \cdot [\binom{2k}{k+1} + \binom{2k}{k} + \binom{2k}{k} + \binom{2k}{k-1}]}{2}$$

$$< \frac{1 + \frac{1}{2^{2k}} \cdot \frac{1}{4} \cdot 4 \cdot \binom{2k}{k}}{2}$$

$$= p_k$$

$\square$

This means that the more vectors we bundle together, the closer $\theta$ is to 90 degrees.

## C  Learning with Group VSA

For a cyclic group VSA model, similar as the binary HDC case, we initialize a linear model with weights $\mathbf{W}$ of size $\#\,\mathrm{class} \times D$, where each element belongs to $G = \mathbb{Z}/n\mathbb{Z} = \{0, 1, \cdots, n-1\}$. Inputs to this classifier are encoded hypervectors $v \in G^D$, the model computes per-class similarities with defined similarity function:

$$\mathcal{S}(x, y) = \langle \psi(x), \psi(y) \rangle = \cos(2\pi(x-y)/n), \forall x, y \in G,$$

which extends to higher dimensional space via

$$\mathcal{S}([x_1, \ldots, x_D], [y_1, \ldots, y_D]) = \frac{1}{D} \sum_{i=1}^{D} \mathcal{S}(x_i, y_i).$$

We calculate the cross-entropy loss between the classifier outputs and the labels, and do back propagation using high precision numbers such as 16-bit floats. At each optimizer step, SGD optimizer moves $\mathbf{W}$ away from $G^{\#\,\mathrm{class} \times D}$, we pull it back with (Fast)-Round operation to finish current step before entering next step. Similar as the binary case, the inference cost remains the same as the bundling method.