# OpenReview forum: "Understanding Hyperdimensional Computing for Parallel Single-Pass Learning"
_NeurIPS.cc/2022/Conference — NeurIPS 2022 Accept_

### Official Review · Reviewer_y6Vm · 2022-07-10

**Rating:** 7
**Confidence:** 2
**Soundness:** 4 excellent
**Presentation:** 4 excellent
**Contribution:** 4 excellent

**Summary:**

The authors present a new measure of learning ability (expressivity) for any vector symbolic architecture (VSA) based on similarity matrices. Specifically, if any VSA (including HDC) cannot express a specific similarity matrix, corresponding to the optimal solution of a learning problem, then the VSA cannot converge to the optimal solution for the learning problem. Using this concept of expressivity, the authors highlight that HDC vectors of any large dimension cannot express certain similarity matrices. Further, they demonstrate that HDC vectors initialized with random Fourier features have higher expressivity than randomly selected ones which in turn results in higher accuracy on MNIST (image-classification task). Lastly, the authors define group VSAs which extends HDC to hypervectors that have much higher expressivity (and accuracy on MNIST/Fashion MNIST image classification tasks).

**Questions:**

Kindly address questions discussed in Strengths and weaknesses section.

**Limitations:**

The authors can make note that the proposed SGD training for HDC and group VSA models might be more energy consuming than existing HDC learning (single pass). Further, the group VSA hardware implementation is notably costlier than binary HDC.

**Strengths And Weaknesses:**

The paper presents a good survey of prior art and highlighting how the results of this paper are novel as they present a theoretical formulation of how to assess the limitations of HDC hypervectors compared to other VSAs.

The authors do a good job of defining all functions/params which helps improve the readability of the paper despite the detailed mathematical discussion.

Section 8 shows good results confirming the expectation that RFF improves HDC performance while the group VSA’s continue to improve it further. It is interesting that the accuracy improvement achieved with group VSA hypervectors is much smaller, as I would have expected otherwise considering the much higher learnability of group VSAs. It would be interesting if the authors can add data for another learning task (aside from the Bayes-optimal classifier) where HDC (RFF/otherwise) is unable to learn the optimal result or has a much poorer accuracy.

Section 8.1 tries to assess the hardware cost of implementing the different HDC/VSA approaches. For binary HDC, the authors select a ripple-carry adder, it’s unclear why they didn’t choose parallel adder circuits like carry select adder which has lower depth and could have made the HDC implementation further cheaper. While CDC metric captures the latency and hence performance aspect of the cost, measuring the change in number of operations might be a more realistic estimate of the energy/area penalty incurred on actual hardware. Since higher complexity group VSA’s will require a much larger number of ops for small accuracy improvements for certain tasks.

 It is useful that the authors have plotted the accuracy vs CDC tradeoff in Figure 1, I was curious if combining with efficient hardware design approaches like Basaklar et. al., [2021], could help lower the CDC for both HDC and group VSAs by reducing the number of elements in each hypervectors (currently 10k). This in turn might help establish group VSA’s as more accurate for a given hardware cost.

---

> ### Author Response · Authors · 2022-08-02
> **Author Response**
>
> We thanks the reviewer for regarding this paper as novel and highly readable with a good survey of prior work, please see below for answers to the questions:
>
> -- Q1: “accuracy improvement achieved with group VSA hypervectors is much smaller … ”
>
> The required similarity matrix to learn (some) tasks in the paper might already be covered by (or close to in terms of Frobenius norm) the set of expressible similarity matrices of the 10k dimensional HDC. Hence, the improvement from group-VSA is limited. If instead considering a 1k dimensional binary HDC with a smaller set of expressible similarity matrices, group-VSA demonstrates a much better accuracy improvement. Our new experiments show that 1k-dimensional binary HDC achieves ​​65.59% 10-epoch test accuracy on MNIST. G(2^3)-VSA, meanwhile, achieves 88.61%, and G(2^4)-VSA achieves 92.56% test accuracy. We plan to include datasets with more variety and complexity in the future.
>
> -- Q2: why not choose parallel adder circuits which has lower depth and could have made the HDC implementation further cheaper?
>
> Yes, other efficient adder implementations will make the CDC of HDC even better, but we chose the ripple carry adder because this serves as a preliminary analysis on the complexity, similar to the big O notation. We do not want to be over-optimistic on the estimation. For practical hardware implementation, a better adder should definitely be used.
>
> -- Q3: measuring the change in number of operations might be a more realistic estimate of the energy/area penalty?
>
> Yes, that’s true. CDC is more of an estimation on latency. We will carefully estimate energy/area based on the number of operations (and maybe other hardware target specific properties) in the future work.
>
> -- Q4: combining with efficient hardware design approaches like Basaklar et. al., [2021] could help lower the CDC for both HDC and group VSAs?
>
> This is a good suggestion. There might be a chance of combining this work with other efficient hardware approaches and getting benefits from both, e.g., first initialize hypervectors using the proposed RFF, then refine them using the approach in Basaklar et. al., [2021].
>
> For the limitations, we will add in the text that SGD training is more costly, though it does not affect inference performance. Group VSA implementation is also slightly more costly as analyzed by CDC.

---

### Official Review · Reviewer_s9Zr · 2022-07-11

**Rating:** 7
**Confidence:** 3
**Soundness:** 3 good
**Presentation:** 4 excellent
**Contribution:** 3 good

**Summary:**

Hyperdimensional computing is an emerging learning paradigm with two main thrusts attempting to improve its accuracy to match SOTA (DL) methods: the first focussed on longer binary vectors and the second on increasing complexity of vector elements (e.g. floating point, complex numbers). The second is often referred as Vector Symbolic Architecture (VSA). The paper introduces the notion of expressivity of VSA as the similarity matrices that can be represented (this is closely linked to learnability). It argues that VSA with more complex hypervectors is more expressive than HDC with binary hypervectors of any dimension.  It proposes a new initialization of binary hypervectors using random Fourier features.

**Questions:**

Are there other notions of similarity different from inner product that could yield different results with respect to expressiveness of the hypervectors or is the result independent of how we define similarity?

Lemma 4.1 is interesting but could authors provide an intuition for the class of matrices that binary HDC cannot express?

**Limitations:**

There are no potential negative societal impact of this work to the best of this reviewer's understanding.

**Strengths And Weaknesses:**

+ The paper defines a notion of expressiveness using similarity matrices and studies the expressiveness of HDC with complex hypervectors vs longer hypervectors.

+  The proposes a new initialization of binary hypervectors using random Fourier features, and shows an improvement by 6.4% on MNIST classification.

+ A new group VSA is defined which is more complex than binary hypervectors but less than floating-point. This improves MNIST accuracy by an addition 1.2%.

Statement 4.1 is a bit confusing. The second statement says that any VSA that can express M can learn this task - that seems to be tautological (if no VSA can express M, the statement will hold vacuously). It would be better to characterize some VSA can express M and learn it.

---

> ### Author Response · Authors · 2022-08-02
> **Author Response**
>
> Thanks for reviewing the paper and providing thoughtful suggestions and summary to our work, please see below for answers to the questions:
>
> -- For question in the Weakness part: “Statement 4.1 is a bit confusing. … ”
>
> Answer:
> In Lemma 4.1, M is a matrix with 1 in the diagonal and -1/2 for the rest entries. Any real/complex VSAs (e.g., unit cycle VSA) can express M in Lemma 4.1. More broadly, any VSA that contains 3 hypervectors where any two of them has -1/2 similarity (120 degree in the inner product sense) will be able to express M. We will make this clearer in the paper.
>
> -- For listed Questions:
>
> Q1: “Are there other notions of similarity … ”
>
> Our similarity expressiveness theory holds for any similarity that can be extensible to an inner product in some space (refer to Definition 2, 4th item), including commonly used metrics such as cosine similarity and hamming distance.
>
> Q2: “intuition for the class of matrices that binary HDC cannot express?”
>
> The set of expressible similarity matrices of binary HDC is the convex hull of the similarity matrices that can be expressed by 1-dimensional binary HDC. In the 3-entity case, it’s the convex hull of 4 basic similarity matrices enumerated in the proof of Lemma 4.1, hence, the class of matrices that binary HDC cannot express are matrices not in the convex hull, please check the beginning section in appendix for more details.

---

> > ### Comment · Reviewer_s9Zr · 2022-08-10
> > **Thank you**
> >
> > The main concerns of the reviewer have been addressed. The reviewer is raising the score.

---

### Official Review · Reviewer_tDsX · 2022-07-11

**Rating:** 7
**Confidence:** 2
**Soundness:** 4 excellent
**Presentation:** 4 excellent
**Contribution:** 3 good

**Summary:**

The authors present a generalization of hyperdimensional computing which supports arbitrary finite groups as elements. They discuss representational power of classical VSAs, propose a novel initialization mechanism, describe their generalization both abstractly and with a concrete example (cyclic groups), and present accuracy results.

**Questions:**

(1) You describe the alternative conventional approach to improving accuracy as increasing $D$ (L35). Given that the space required for a group VSA scales with its group size and its CDC only logarithmically, it seems to imply that comparing a 10K-long HDC model with a 10K-long G($2^3$)-VSA model is giving the group VSA model an 8x advantage in resources (at vaguely similar CDC). What is the accuracy of an HDC with D of $2^3\times 10000$ versus a 10K-long G($2^3$)-VSA? Same question for $2^4$.

(2) Given the (welcomed) emphasis on establishing the unit cycle VSA within your group VSA generalization, I was a bit surprised not to see it in the results. I assume there's a good reason for this (i.e., is compute time absurd?), but given its prominence in the rest of the paper, I was wondering what that reason is.

**Limitations:**

(Addressed in other sections.)

**Strengths And Weaknesses:**

**Originality**

The paper's three main contributions (an anaylsis of expressivity of binary HDC, an approach to hypervector construction, and a generalized model of VSAs) all seem novel, somewhat orthogonal (and thus useful even in the absence of the other two), and applicable. The use of RFF for HDC is a meaningful adaptation of the idea, and group VSA (especially its concrete form over $\mathbb{Z}/n\mathbb{Z}$) is a clear and helpful extension of binary HDC.

**Quality**

The analysis and development of RFF initialization and group VSA seems solid, well-motivated, and self-contained. The authors have struck a good balance between capturing high-level concepts (motivation by similarity matrices in Def 1; arbitrary VSAs in Def 2) and concrete applications of them (counterexample $\textrm{M}_{\textrm{Lemma4.1}}$; cyclic group VSAs in 6.2). I appreciated having both aspects developed in parallel.

Section 7 seemed overly brief. While the motivating problem seems clear enough, though perhaps it would've been helpful to understand the efficacy of the approach in avoiding the problem and/or the effect of number of epochs.

The results seem to suggest that nearly all of the accuracy benefits stem from RFF initialization. The practical benefits of group VSA seem limited, if not a bit mixed. I still think the contribution of a generalized formulation of group VSA is useful regardless, but it would have been nice to see some discussion as to why the extra representational power doesn't translate to much better accuracy.

The CDC analysis was a welcomed addition. Given the importance of an efficient hardware implementation as a motivation for the existence of HDC, I appreciated the nod to fleshing out the proposal of group VSA with the impact to its implementation.

**Clarity**

The writing is clear, and most of the paper flows well from one section to another. Analysis leads naturally to proposals, and most sections progress from abstract to concrete in a clean and understandable way.

**Significance**

This paper seems like a solid addition to the body of work on HDC. Its contributions each stand on their own and form a solid foundation for further work. It seems easy to build off, self-contained and methodical in its approach, and useful for understanding the behavior of HDC.

---

> ### Author Response · Authors · 2022-08-02
> **Author Response**
>
> We thanks the reviewer for acknowledging our main contributions as novel and solid to a better understanding of hyperdimensional computing, please see below for answers to the questions:
>
> -- For question in the Quality part: why the extra representational power doesn't translate to much better accuracy？
>
> Answer:
> The required similarity matrix to learn (some) tasks in the paper might already be covered by (or close to in terms of Frobenius norm) the set of expressible similarity matrices of the 10k dimensional HDC. Hence, the improvement from group-VSA is limited. If instead considering a 1k dimensional binary HDC with a smaller set of expressible similarity matrices, group-VSA demonstrates a much better accuracy improvement. Our new experiments show that 1k-dimensional binary HDC achieves ​​65.59% 10-epoch test accuracy on MNIST. G(2^3)-VSA, meanwhile, achieves 88.61%, and G(2^4)-VSA achieves 92.56% test accuracy.
>
> -- For listed Questions:
>
> Q1: At the same hypervector dimensionality, G(2^3)-VSA model has a 3x (not 8x) model size advantage as compared to a binary HDC since 3 bits are sufficient to store each vector element. In terms of energy/area, it is more difficult to analyze the advantage because it depends on a more concrete hardware implementation (e.g., which type of adder to use). Hence, it is hard to conduct a fair comparison between a 30k (not 80k) binary HDC and a G(2^3)-VSA. We may leave a more comprehensive analysis to the future work.
>
> For the sake of time, if we only focus on the model size and consider an HDC/VSA whose dimensionality is in the thousands, our new experiments show that the 1k-dimensional G(2^3)-VSA achieves a 1.12% better accuracy compared to 3k-dimensional binary HDC on MNIST. Similarly, 1k-dimensional G(2^4)-VSA has a 1.26% better accuracy than 4k-dimensional binary HDC. See below: 3k-dimensional HDC: 87.49%, 4k-dimensional HDC: 91.30%, 1k-dimensional G(2^3)-VSA: 88.61, 1k-dimensional G(2^4)-VSA: 92.56%.
>
> Q2: The major reason is hardware efficiency. Given the results in Figure 1 where higher orders beyond group of 4 do not seem to improve the model accuracy any more on the given benchmarks, we decide to stop increasing the group order to get a balance between model complexity and accuracy. The unit circle VSA is indeed a generalized version within the proposed group VSA, so for completeness we include it. Practically it also requires floating-point arithmetics that are much more expensive in terms of latency and power, so it is a trade-off whether or not to extend to that point.
>
> We would like to thank again for the thoughtful suggestions, section 7 is brief mainly due to the space constraints. A comprehensive study on how the proposed learning approach outperforms bundling is definitely helpful, though it is beyond the scope of this work as we are more focused on analyzing the limits of HDC. We will mark it on future TODOs.

---

### Official Review · Reviewer_RVTb · 2022-07-12

**Rating:** 7
**Confidence:** 3
**Soundness:** 4 excellent
**Presentation:** 3 good
**Contribution:** 4 excellent

**Summary:**

The authors present an analysis on vector symbolic architectures (VSAs) applied to canonical classification tasks. They comment on the variety of approaches in this space and provide some theoretical arguments for the limitations of hyperdimensional computing (HDC), a binary VSA. Finally, they propose a new form of VSA that maintains the desirable hardware compatibility while improving classification performance.

**Questions:**

1) I was confused about the methodology used for Table 1. Were the 1-epoch results for the proposed VSA models trained using SGD or bundling? Do you have a comparison for these two training methods?

2) I think it would be valuable if the authors investigated the observation that VSA can perform better with a single pass of the training set than with multiple passes. Specifically, I think more effort should be made to convince the reader that this is a benefit of the model and not a shortcoming of the chosen optimization or model design process (e.g. a result of overfitting).

3) I am curious to hear the authors’ perspective on how the similarity matrix relates to the classical outer-product memory used in Hopfield networks.

4) typo on line 364 “VSA stikes a good trade”


**Limitations:**

The authors describe trade-offs for the various VSA algorithms throughout the text, although there is no explicit section naming limitations to their current approach. This would be a valuable addition to the paper.

**Strengths And Weaknesses:**

**Originality:** As far as I know the proposed group VSA & using the similarity matrix as an analysis of expressive capacity are novel contributions.

**Quality:** The methods used are appropriate for their questions. I did not read the appendix to verify the claims.

**Clarity:** The authors provided a pedagogical overview that covered the key ideas necessary to understand their contribution.

**Significance:** I believe this work is a significant contribution towards reducing the exorbitant energy costs of popular classification algorithms (e.g. DNNs).

---

> ### Author Response · Authors · 2022-08-02
> **Author Response**
>
> We thanks the reviewer for considering our work as novel and significant contributions to the area, please see below for answers to the questions:
>
> Q1: All results in Table 1 are trained using SGD (fair comparison). We didn’t train using the bundling method because of the analysis in Section 7 — the class representatives obtained from bundling will become unreliable (no longer representatives) when the dataset size is large, e.g., 60k images in MNIST.
>
> Q2: VSA with a single pass can perform reasonably well. As a matter of fact, with multiple passes, VSA is doing slightly better than single pass in most cases with the only exception on Fashion-MNIST with RFF-G(2^4)-VSA. We will investigate this aspect in future work.
>
> Q3: We currently do not have a good understanding of Hopfields networks. It’s not clear to us how to connect the outer-product memory with the similarity matrix. But from our superficial understanding, in the HDC/VSA framework, each entry of the similarity matrix is the similarity score of two hypervectors, measured by inner product. It can be easily connected to the learnability of a task since the similarity score is used in inference. In the Hopfield network, the outer-product memory is the sum of the self outer-product matrix of each “pattern”, which seems not related to similarity measurement.
>
> Q4: we will correct the typo as suggested.
>
> Thanks for the suggestion on an explicit trade-offs section, we will try to add this text in the future version.

---

> > ### Comment · Reviewer_RVTb · 2022-08-10
> > **reply to author rebuttal**
> >
> > Thank you for addressing my concerns. I have no further questions or comments.

---

### Meta-Review · Area_Chair_EmWe · 2022-08-26

**Recommendation:** Accept
**Confidence:** Certain

**Metareview:**

This paper analyzes some existing limitations of hyperdimensional computing (HDC), and proposes new techniques to alleviate them, showing improvements in model quality while maintaining hardware efficiency.

There is a strong consensus among all reviewers that this is a solid submission whose contributions are novel, insightful, and likely to inspire future work in this important direction. Reviewers's questions and concerned were adequately addressed by the authors during the discussion period, making it a clear "Accept".

**Award:**

No

---

### Decision · Program_Chairs · 2022-09-14

Accept